# Structural and Functional Changes of Reconstituted High-Density Lipoprotein (HDL) by Incorporation of α-synuclein: A Potent Antioxidant and Anti-Glycation Activity of α-synuclein and apoA-I in HDL at High Molar Ratio of α-synuclein

**DOI:** 10.3390/molecules26247485

**Published:** 2021-12-10

**Authors:** Kyung-Hyun Cho

**Affiliations:** 1LipoLab, Department of Medical Biotechnology, Yeungnam University, Gyeongsan 38541, Korea; chok@yu.ac.kr; Tel.: +82-53-964-1990; Fax: +82-53-965-1992; 2Korea Research Institute of Lipoproteins, Medical Innovation Complex, Daegu 41061, Korea

**Keywords:** α-synuclein, apolipoprotein A-I, Parkinson’s disease, high-density lipoproteins, oxidation, glycation

## Abstract

α-synuclein (α-syn) is a major culprit of Parkinson’s disease (PD), although lipoprotein metabolism is very important in the pathogenesis of PD. α-syn was expressed and purified using the pET30a expression vector from an *E. coli* expression system to elucidate the physiological effects of α-syn on lipoprotein metabolism. The human α-syn protein (140 amino acids) with His-tag (8 amino acids) was expressed and purified to at least 95% purity. Isoelectric focusing gel electrophoresis showed that the isoelectric point (pI) of α-syn and apoA-I were pI = 4.5 and pI = 6.4, respectively. The lipid-free α-syn showed almost no phospholipid-binding ability, while apoA-I showed rapid binding ability with a half-time (T_1/2_) = 8 ± 0.7 min. The α-syn and apoA-I could be incorporated into the reconstituted HDL (rHDL, molar ratio 95:5:1:1, palmitoyl-2-oleoyl-sn-glycero-3-phosphocholine (POPC):cholesterol:apoA-I:α-syn with the production of larger particles (92 Å) than apoA-I-rHDL (86 and 78 Å) and α-syn-rHDL (65 Å). An rHDL containing both apoA-I and α-syn showed lower α-helicity around 45% with a red shift of the Trp wavelength maximum fluorescence (WMF) from 339 nm, while apoA-I-HDL showed 76% α-helicity and 337 nm of WMF. The denaturation by urea addition showed that the incorporation of α-syn in rHDL caused a larger increase in the WMF than apoA-I-rHDL, suggesting that the destabilization of the secondary structure of apoA-I by the addition of α-syn. On the other hand, the addition of α-syn induced two-times higher resistance to rHDL glycation at apoA-I:α-syn molar ratios of 1:1 and 1:2. Interestingly, low α-syn in rHDL concentrations, molar ratio of 1:0.5 (apoA-I:α-syn), did not prevent glycation with more multimerization of apoA-I. In the lipid-free and lipid-bound state, α-syn showed more potent antioxidant activity than apoA-I against cupric ion-mediated LDL oxidation. On the other hand, microinjection of α-syn (final 2 μM) resulted in 10% less survival of zebrafish embryos than apoA-I. A subcutaneous injection of α-syn (final 34 μM) resulted in less tail fin regeneration than apoA-I. Interestingly, incorporation of α-syn at a low molar ratio (apoA-I:α-syn, 1:0.5) in rHDL resulted destabilization of the secondary structure and impairment of apoA-I functionality via more oxidation and glycation. However, at a higher molar ratio of α-syn in rHDL (apoA-I:α-syn = 1:1 or 1:2) exhibited potent antioxidant and anti-glycation activity without aggregation. In conclusion, there might be a critical concentration of α-syn and apoA-I in HDL-like complex to prevent the aggregation of apoA-I via structural and functional enhancement.

## 1. Introduction

Parkinson’s disease (PD), which is the second most common neurodegenerative disease in the elderly population, is frequently associated with mutations of α-synuclein (α-syn) [1,2]. PD involves the loss of dopaminergic neurons in the substantia nigra of the midbrain, and the presence of protein aggregates, i.e., Lewy bodies, involving α-syn [3]. Brain cholesterol and lipoprotein are believed to play an important role in the action of α-syn, a major component of Lewy bodies, in the pathogenesis of PD [4]. Low levels of serum apolipoprotein A-I (apoA-I) and HDL-C are correlated with the earlier onset of PD [5,6]. Lower plasma HDL and apoA-I levels have been associated with earlier PD onset [7] and higher PD risk [8,9,10].

While LDL-cholesterol is a major carrier of blood cholesterol, brain cholesterol handling is strongly dependent on the high-density lipoprotein (HDL) metabolism, and HDL-like particles can be synthesized from the brain and glial cells [11]. In the brain, there are no LDL-like particles and apo-B containing lipoproteins as cholesterol carriers [12]. As with apoA-I, α-syn can also bind cholesterol as an acceptor from ABCA1-mediated cholesterol efflux [13,14]. Serum HDL can interact with α-syn in the brain [15]. Therefore, a putative interaction between α-syn and apoA-I might frequently occur to regulate cholesterol homeostasis. On the other hand, there is limited information on α-syn and HDL in the protein level on the neuron system.

The functionality of HDL is essential for suppressing the incidence of diabetes, cardiovascular disease, and stroke [16,17]. Structural and functional correlations of HDL are highly dependent on apoA-I because apoA-I is major protein component of HDL [18]. ApoA-I, 243 amino acids in mature sequence, exerts anti-oxidant and anti-inflammatory activity with tandem repeats of 11 and 22 amino acid helix domain [19]. Interestingly, there is a high sequence homology between α-syn and apoA-I, and they have several tandem amphipathic helices.

A nationwide cohort study showed that many risk factors of metabolic syndrome, such as hypertension, diabetes mellitus, and dyslipidemia, are also important risk factors of PD [20]. In addition to Alzheimer’s disease, glycated α-syn oligomers activate microglia and induce the release of cytokines and NF-kB signaling proteins, which are linked with neuroinflammation and neuronal cell death in PD [21]. The glycation of α-syn is a factor that affects the aggregation of the protein into LB in PD. Glycation was first reported in the substantia nigra and locus coeruleus in the periphery of LB [22].

A hypothesis to explain the initial pathogenic mechanism of PD, viral infection in the CNS, is associated with the incidence of PD via neuroinflammation, synaptic dysfunction, and autophagy disruption [23,24]. Paraoxonase (PON-1) is potent anti-oxidant enzyme, which is associated with HDL, and displayed that higher PON-1 levels suppressed risk of PD [25]. Patients with PD also showed various genetic polymorphisms of the PON-1 gene and enzyme activity [26], which is responsible for anti-viral activity. Regarding the anti-infection ability, the glycation of HDL impaired its anti-inflammatory activity in the innate immunity against viral infection through the loss of (PON-1) activity [27]. It has been well known that viral infection is associated with elevation of PD risk and possible links between COVID-19 and PD is now emerged [28]. Native HDL with higher PON-1 activity showed potent antiviral ability against SARS-CoV-2, while glycated HDL with lower PON-1 activity lost its antiviral activity [27].

Because α-syn can be detected in the saliva, cerebrospinal fluid, and blood as a biomarker of PD [29], there is a strong possibility that α-syn can interact with other proteins in the blood and brain. More than 99% of the α-synuclein in human blood is present in the peripheral blood cells, with the remainder in plasma. Blood α-syn level is associated with PON-1 genotype and environmental factors [30]. Although α-syn can bind to cholesterol, the role of α-syn in the lipoprotein metabolism is largely unknown, particularly in the interaction with apoA-I. ApoA-I is found in the brain and is delivered mainly from the blood by crossing the blood-brain barrier (BBB), while apo-E in the blood cannot cross the BBB [31].

No study has examined the physicochemical characters of α-syn regarding the interaction of lipid and apoA-I. This study examined the mutual interaction of apoA-I and α-syn on the purified protein level in the lipid-free and lipid-bound state. In the current study, human α-syn was expressed, purified, and characterized to elucidate the physiological effects of α-syn in lipoprotein binding with apoA-I in the presence of oxidation and glycation stress.

## 2. Results

### 2.1. Purification and Characterization of α-syn

The α-syn band was detected as a single band at least 95% purity of approximately 20 kDa band position from 15% SDS-PAGE (Figure 1A). The isoelectric point (pI) was detected as approximately 4.5–4.7 from isoelectric focusing (Figure 1B). The calculated molecular weight of α-syn was based on 148 amino acid sequence is 15525.3 Da with pI = 5.1. Protein prediction analysis revealed 3% of the β-strand and 39% of the coiled loop structure. The extinction coefficient at 280 nm (ε_280_) was measured as 5781 M/cm based on UV spectroscopy using a DU800 spectrophotometer (Beckman, Palo Alto, CA, USA) and a Suprasil quartz cuvette (1-cm path length).

### 2.2. Phospholipid Binding Ability

The addition of α-syn had almost no phospholipid (dimyristoylphosphatidylcholine, DMPC) clearance activity, while apoA-I showed rapid binding ability, as shown in Figure 2. Native apoA-I alone showed a half-time of removal (T_1/2_) of 7 ± 1 min, but α-syn showed only 13% of DMPC clearance from the initial level during 60 min.

### 2.3. Synthesis of rHDL

In the non-denaturing state, lipid-free α-syn showed three major bands at approximately 65 Å, 73 Å, and at the bottom of the gel, indicated by the blue arrow (lane 2), as shown in Figure 3A. In contrast, apoA-I (lane 1, Figure 3A) showed a typical broadband pattern from 63, 69, 71 to 73 Å, as indicated by the yellow arrowhead. A mixture of α-syn and apoA-I (lane 3) showed a similar pattern of α-syn alone. After rHDL synthesis, rHDL-(α-syn) showed a single band around 65 Å, while apoA-I-rHDL showed 86 and 78 Å. A mixture of α-syn and apoA-I resulted in five bands from 92, 80, 74, 69, and 65 Å with a smeared band intensity. These results indicate that two major bands α-syn in the lipid-free state were combined in one particle of α-syn-rHDL, while a mixture of apoA-I and α-syn produced more heterogeneous bands in the rHDL state.

### 2.4. Secondary Structure Analysis

In the lipid-free state, as shown in Figure 4A, apoA-I-rHDL showed the typical CD spectra of the α-helix with two minima at 209 nm and 222 nm, while α-syn showed a negative CD band with the lowest ellipticity (−29.842) at 195 nm, which was characteristic of the disordered structure. On the other hand, a mixture of apoA-I and α-syn showed two minima at 208 nm and 222 nm for the α-helix spectrum (Figure 4A). In the rHDL state, as shown in Figure 4B, apoA-I-rHDL showed larger ellipticity of the CD spectrum with two minima 209 and 222 nm. α-syn showed remarkably increased ellipticity (−17.488) with a negative CD band at approximately 198 nm, which was characteristic of the disordered structure. A mixture of apoA-I and α-syn showed a typical α-helical spectrum with two minima at 208 and 222 nm. A calculation of the α-helix content from an ellipticity of 222 nm revealed that apoA-I alone and a mixture of apoA-I and α-syn showed 56% and 51% of α-helicity, respectively, in the lipid-free state (Table 1). In the rHDL state, apoA-I alone and a mixture of apoA-I and α-syn showed 76% and 45% α-helicity, respectively, suggesting that the incorporation of α-syn caused a larger decrease in α-helicity in the rHDL state than the lipid-free state (Table 1).

In the lipid-free state, apoA-I and a mixture of apoA-I and α-syn showed 341 nm and 332 nm of wavelengths of maximum fluorescence (WMF), respectively, suggesting that the incorporation of α-syn caused a blue shift of Trp in apoA-I. In the rHDL state, apoA-I showed 337 nm of WMF, while a mixture of apoA-I and α-syn showed 338 nm (Table 1). This result suggests that the addition of α-syn caused a red-shift of WMF, which showed different behavior of α-syn between the lipid-free and lipid-bound state. The incorporation of α-syn in rHDL destabilized the α-helical structure in apoA-I with a red-shift of Trp, which contributed to the destabilization of apoA-I in rHDL.

### 2.5. α-syn Caused More Rapid Isothermal Denaturation

The addition of urea caused the denaturation of apoA-I in the lipid-free and lipid-bound state, as shown in Figure 5. At the initial level, lipid-free apoA-I and rHDL showed a WMF of approximately 337–339 nm. The lipid-free apoA-I initiated denaturation process with a 4 nm increase in WMF by 1 M urea addition, while apoA-I in the rHDL state was more resistant to denaturation until the addition of 4 M urea around 340 nm of WMF. On the other hand, the addition of α-syn in rHDL caused a more rapid increase in WMF from 1 M urea addition to a WMF of 341 nm. Until 7 M urea addition, rHDL(apoA-I + α-syn) showed a 2–4 nm higher WMF than rHDL(apoA-I alone), indicating that the destabilization of HDL was accelerated by α-syn addition.

### 2.6. Inhibition of Glycation by α-syn

The fructose treatment resulted in severe glycation of apoA-I with an 18-fold increase in yellowish fluorescence from the initial level (Figure 6A) and the multimerization of the apoA-I band (lane 2, Figure 6B) during 144 h incubation. The addition of α-syn at a low molar ratio (apoA-I:α-syn = 1:0.5) also showed a severe extent of glycation with the multimerization of apoA-I (lane 4). Interestingly, the addition of α-syn in a higher molar ratio (apoA-I:α-syn = 1:1 and 1:2) resulted in a small extent of glycation with only an approximately five-fold increase in WMF from the initial level without multimerization of apoA-I (lanes 5 and 6). These results suggest that a putative anti-glycation activity of α-syn was dependent on the concentration in the apoA-I mixture.

In a mixture of apoA-I and α-syn, a multimerized band was detected only in a low α-syn content (apoA-I:α-syn = 1:0.5) under a fructose treatment (Figure 7A). With the increase in α-syn content, there was no multimeric band in Coomassie Brilliant Blue staining (lanes 3 and 4). Western blot with the apoA-I antibody also showed that the multimerization occurred only in a low α-syn content (apoA-I:α-syn = 1:0.5) up to the tetrameric apoA-I. On the other hand, the multimerization was inhibited by the addition of α-syn in a dose-dependent manner. No multimeric band was observed in the highest of α-syn content (apoA-I:α-syn = 1:2), as shown in lane 4 of Figure 7B.

### 2.7. Inhibition of LDL Oxidation

The treatment with cupric ions caused LDL oxidation with the largest increase in absorbance at 234 nm (A_234_), approximately 3.4-fold from the initial level, while native LDL produced almost no increase in A_234_ during 110 min incubation (Figure 8). The lipid-free apoA-I showed potent antioxidant activity to suppress LDL oxidation, an approximately 2.3-fold increase in A_234_ from the initial level. Interestingly, α-syn exhibited stronger inhibitory activity to suppress LDL oxidation than apoA-I, around a 1.5-fold increase in A_234_ from the initial level (Figure 8). A mixture of apoA-I and α-syn showed sufficient inhibitory activity against LDL oxidation around a 1.8-fold increase in A_234_. To the best of the authors’ knowledge, this is the first report to show that α-syn possess the potent inhibitory activity of LDL oxidation.

In lipid-bound state, rHDL(α-syn) also showed stronger inhibitory activity than rHDL(apoA-I) to suppress cupric ion-mediated LDL oxidation by up to 30% (Figure 9). Interestingly, rHDL(apoA-I + α-syn) showed the strongest antioxidant activity until 80 min, indicating that the synergistic effects between apoA-I and α-syn in rHDL.

LDL oxidation was accelerated more by the fructose treatment than cupric ion alone, especially in the early phase, until 40 min incubation, as shown in Figure 10. The fructose treatment caused the highest production of A_234_ in the presence of apoA-I and cupric ion, suggesting that oxidation and glycation caused a synergistic contribution, increasing the production of conjugated diene up to a 2.4-fold increase in A_234_. Interestingly, the addition of α-syn in rHDL caused the potent inhibition against the LDL oxidation, particularly in a high molar ratio of 1:1 and 1:2, resulting in almost no conjugated diene production. On the other hand, the low α-syn content in rHDL (apoA-I:α-syn = 1:0.5) lost the inhibition, resulting in a 1.4-fold increase in A_234_ from the initial level.

In agarose gel, the treatment of Cu^2+^ and fructose (lanes 2 and 4) caused the fastest electromobility of LDL because more oxidized LDL moved faster to the bottom of the gel. The addition of α-syn, particularly at high molar ratios, caused the slowest electromobility of LDL (lanes 5, 7, and 8 in Figure 10B) almost the same band position with native LDL (lane 1), supporting the potent antioxidant activity of α-syn. On the other hand, the low content of α-syn in rHDL, 1:0.5 (apoA-I: α-syn), showed the loss of antioxidant ability with faster electromobility (lane 6). These results suggest there might be a critical point to cause the destabilization of apoA-I and α-syn in the rHDL.

### 2.8. Embryo Survivability by Microinjection

The same amount of each apoA-I and α-syn (final 5 μM) was injected into zebrafish embryo at 2 h post fertilization to test the possible positive or negative effect in embryo development. During 48 h incubation, an apoA-I injected embryo showed significantly higher survivability of approximately 74% than the α-syn-injected embryo (approximately 68% survivability), while PBS injected embryo showed similar survivability with apoA-I around 78% survivability (Figure 11A). Stereoscopic observations showed that α-syn injected embryo showed a slower developmental speed than that shown in Figure 11B. Visualization of ROS by DHE staining showed that the α-syn injected embryo showed slightly higher ROS production than the apoA-I injected embryo.

## 3. Discussion

The putative roles of α-syn in aggregation to form Lewy bodies remain to be investigated, even though it is very important to elucidate the initial mechanism of PD pathogenesis [32]. Because α-syn shows binding ability with lipid using its conserved amphipathic helix domain, it can be considered as an apolipoprotein [33]. On the other hand, despite these similarities, the structural and functional correlations of α-syn regarding the HDL-like macrostructure have not been studied sufficiently.

In regarding a possible interaction between α-syn and HDL-associated proteins, it has been known that apo-J/clusterin prevented α–syn aggregation due to its chaperone activity [34]. Apo-D also possesses neuroprotective effects against oxidative stress in glial cell of the substantia nigra [34]. It is plausible that apo-E also involves in PD, because many brain tissues synthesize its apo-E to maintain cholesterol homeostasis via the production of HDL-like molecules. An increased apo-E level is also a risk factor of PD via α-synucleinopathies [35]. Although the mechanism is not fully elucidated, apo-E isoforms independently regulate α-syn pathology and contribute to disease progression in the synucleinopathies [36]. The isoform of apoEε4 increases α-syn aggregation more than other isoforms [37], particularly at lower concentrations, even though the precise mechanism is unknown.

Nevertheless, the effects of apoA-I during the α-syn aggregation process are not completely understood. This study was designed to provide information on the physiological effects of α-syn upon the binding of a phospholipid, cholesterol, and apoA-I to reconstitute HDL. A recombinant α-syn was expressed and purified (Figure 1), and characterized using a biochemical assay, such as phospholipid-binding ability (Figure 2), synthesis of reconstituted HDL (Figure 3), and secondary structure (Figure 4) with isothermal denaturation (Figure 5). Interestingly, α-syn could not bind with phospholipid (DMPC) in the absence of apoA-I but bind with phospholipid (POPC) in the presence of apoA-I. The highly disordered structure of α-syn, which has less than 1% α-helicity, was changed into an ordered structure by the addition of apoA-I and changed into an α-helical conformation both in the lipid-free and lipid-bound state, as shown in Figure 4. Similarly, α-syn in 1,2-dioleoyl-sn-glycero-3-phospho-L-serine (DOPS) lipoprotein particles showed a typical α-helical CD spectrum with two minima peak at 210 and 221 nm and one maxima peak at 195 nm [38]. Overall, these results suggest that α-syn has considerable structural plasticity upon specific phospholipid binding, especially in apoA-I.

The nonenzymatic glycation of protein is associated with an increased risk of PD via the accumulation of advanced glycated end (AGE) products [39]. The glycation of α-syn is linked to the formation of insoluble protein plaque and toxic oligomers that can cause neuronal death. The AGE are detected in LB depositions in PD patients. In the development of PD, α-syn aggregation is an important pathophysiological mechanism and AGE crosslinks the α-syn aggregation [40]. Glycation of apoA-I is also a hallmark of diabetic complications to exacerbate neurodegenerative diseases via acceleration of amyloidogenesis. On the other hand, no study has compared the glycation susceptibility of apoA-I and α-syn. The current study showed that α-syn exhibited potent anti-glycation activity at high apoA-I:α-syn molar ratios (1:1; 1 mg: 0.55 mg, corresponding final 35 μM), in the lipid-free state (Figure 6 and Figure 7) and rHDL state (Figure 10), while α-syn lost the anti-glycation activity at low apoA-I: α-syn ratios (1:0.5; corresponding 17 μM of α-syn). Interestingly, α-syn at high contents exerted potent anti-glycation activity α-syn and apoA-I having 15 and 21 Lys residues, respectively, as many potential glycation sites. AGE has been reported to be co-localized with α-syn, where they were linked to the accelerated aggregation of the protein [41]. To the best of the authors’ knowledge, this is the first report to show that α-syn exerts potent anti-glycation activity to inhibit the fructosylation of apoA-I. A critical concentration of α-syn might exist around 17–35 μM to inhibit glycation, even although the mechanism remains to be determined.

Moreover, α-syn (final 2 μM) exhibited more potent antioxidant activity to inhibit cupric ion mediated LDL oxidation than apoA-I (final 2 μM) in the lipid-free (Figure 8) and lipid-bound state (Figure 9 and Figure 10). This result shows good agreement with previous reports that α-syn can inhibit the oxidation of unsaturated lipid in vesicles, and there is a critical concentration [42]. Oxidation of α-syn is a key event to cause aggregation and fibrillization of α-syn. Therefore, several active antioxidant ingredients have been developed to inhibit the oxidation of α-syn. On the other hand, it would be more desirable that α-syn has its own antioxidant activity. α-syn may exert its antioxidant activity to maintain a healthy neuronal cell status because lipid peroxidation is essential for α-syn induced cell death [43].

On the other hand, α-syn was reported to be a metal-binding protein [44]. α-syn binds to copper and iron in the C-terminal, which can act as the inhibitory activity of cupric ion mediated LDL oxidation. Metals can damage HDL functionality with destabilization. The ferrous ion (Fe^2+^) treatment resulted in the severe proteolytic degradation of HDL, particularly in the presence of fructose, as reported elsewhere [45]. Thus, the metal-binding ability of α-syn can protect apoA-I from aggregation, as shown in Figure 7, and oxidation, as shown in Figure 9 and Figure 10.

The HDL metabolism in the brain is different from that of the blood and other organs. Brain cells can synthesize apo-E to form apo-E enriched HDL (apoE-HDL) but not apoA-I [11]. The serum apoA-I can cross the blood–brain barrier via a putative transporter to form apoA-I-HDL [31] and inhibit amyloid aggregation [46]. Despite this, no study has investigated the in vitro interactions of α-syn and apoA-I in the lipid-free and lipid-bound state and its physiological effect. Interesting, there is high sequence homology between α-syn and apoA-I with several amphipathic helices [47]. On the other hand, there is limited information in the structural and functional correlations of α-syn, particularly the interaction with apoA-I in HDL. Although α-syn lacks a stable three-dimensional structure in water, it can adopt various conformations depending on environmental factors [47,48]. Generally, the aggregation of α-syn is accompanied by a transition from a random coil monomer to toxic oligomers via the formation of fibril with a beta-sheet conformation. In the presence of apoA-I, however, a disordered structure of α-syn was converted to α-helix (Figure 4), indicating that a protective effect for neurons via the inhibition of aggregation of α-syn in the beta-sheet structure was provided by the presence of apoA-I. Indeed, a lower plasma apoA-I level is associated with the incidence of PD, particularly in earlier age onset patients [8]. It is well established that the circulating blood levels of apoA-I may be proportional to the levels of apoA-I in the brain because apoA-I crosses the BBB and has been proposed as a carrier to deliver target drugs to the brain [48]. These results may contribute to develop a new therapeutic tool of PD via enhancement of HDL to maximize binding affinity of α-syn and inhibition of aggregation. Measurement of HDL functionality also could apply to develop a diagnostic method to test interaction ability of HDL and α-syn.

In conclusion, purified α-syn and apoA-I can interact to maintain a solubilized HDL complex form with α-helical conformation. A higher apoA-I:α-syn molar ratio showed stronger anti-glycation and antioxidant activity to inhibit the aggregation of apoA-I, which is beneficial to prevent the initial PD event. These results suggest that the native α-syn showed adequate physiological activity and had beneficial activity in a putative HDL-like complex with apoA-I.

## 4. Materials and Methods

### 4.1. Materials

The cloned gene of α-synuclein (pOBT7, Cat # hMU010188) was obtained from Korea Human Gene Bank (Daejeon, Korea). The pET30a(+) expression vector and *E. coli* BL21 (DE3) were purchased from Novagen (Madison, WI, USA). The restriction enzymes were acquired from New England BioLabs (Beverly, MA, USA). Palmitoyloleoyl phosphatidylcholine (POPC, #850457) and dimyristoylphosphatidylcholine (DMPC, #850345) was supplied by Avanti Polar Lipids (Alabaster, AL, USA). Sodium cholate (#C1254) was procured from Sigma (St Louis, MO, USA).

### 4.2. Expression and Purification of α-synuclein

Human α-synuclein gene (Korean unigene clone IRAU-50-A12) was cloned using polymerase chain reaction (PCR) with the forward primer 5′-ATGGTACATATGGATGTATTC ATGAAAGGAC-3′ and reverse primer 5′-ATGGTACTCGAGGGCTTCAGGTTCGTA GTC-3′) to generate *Nde I* and *Xho I* sites for constructing the expression vector.

The subcloned cDNA of α-syn was inserted into the pET30a expression vector to verify DNA sequencing using a Sequentator (ABI7500, ABI, Foster City, CA, USA). The expressed polypeptide in mature form was 140 amino acids, and His-tag (8 amino acids, L-E-HHHHHH) was attached to the C-terminal and Met in the N-terminal were used as the start codon. The Hig-tagged α-syn gene was expressed and purified using Ni^2+^-nitrilotriacetic acid column chromatography (Peptron, Cat#1103-3, Daejeon, Korea), as described elsewhere [49,50]. The fractions containing α-syn were pooled and dialyzed against buffer containing 10 mM Tris-HCl (pH 8.0) and 10% glycerol. The protein concentration was determined by Bradford assays using bovine serum albumin (BSA) as a standard. The protein purity was initially monitored by SDS-PAGE and Coomassie blue staining as our previous report [50,51].

### 4.3. Protein Sequencing and Isoelectric Focusing

Protein samples for sequencing were electrotransferred onto a PVDF membrane (Immobilon-P) according to the protocol outlined by Matsudaira [51]. The NH_2_-terminal amino acid sequence of the excised band was determined using an Applied Biosystems model 491A sequencer (Foster City, CA, USA) located in the Korea Basic Research Institute (Daejeon, Korea).

Isoelectric focusing (IEF) was carried out using a precasted gel with PhastGel IEF 3–9 (17-0543-01; GE Healthcare) on a PhastSystem (GE Healthcare), as described elsewhere [52]. The protein bands were visualized by PhastGel Blue R (17-0518-01, GE Healthcare) staining. The range of isoelectric point (pI) for the migrated bands was compared with known pI standard proteins, Serva Liquid Mix (Cat. 39212-01, Invitrogen, Carlsbad, CA, USA) containing amyloglucosidase (pI = 3.6), β-Lactoglobin (pI = 5.1), myoglobin (pI = 6.6), and ribonuclease A (pI = 9.3).

### 4.4. Characterization of Secondary Structure by Circular Dichroism

The average α-helix contents of the proteins in the lipid-free and lipid-bound states were measured by circular dichroism (CD) spectroscopy (J-700, Jasco, Tokyo, Japan) as our previous report [50]. Four scans were accumulated and averaged. The α-helical content was calculated from the molar ellipticity at 222 nm [53].

### 4.5. Trp Fluorescence during Isothermal Denaturation

The wavelengths of maximum fluorescence (WMF) of the tryptophan (Trp) residues in apoA-I were determined from the uncorrected spectra using an LS55 spectrofluorometer (Perkin-Elmer, Norwalk, CT, USA), as described previously [54], using the WinLab software package 4.00 (Perkin-Elmer) and a 1-cm path-length Suprasil quartz cuvette (Fisher Scientific, Pittsburgh, PA, USA). The samples were excited at 295 nm to avoid tyrosine fluorescence, and the emission spectra were scanned from 305 to 400 nm at room temperature [55]. For isothermal denaturation, the effects of urea addition on the secondary structures and apoA-I in a lipid-bound state were monitored by measuring tryptophan movement using fluorospectroscopy, as reported elsewhere [56].

### 4.6. Purification of Human Lipoproteins

Human LDL (1.019 < d < 1.063) and HDL (1.063 < d < 1.225) were isolated from the sera of young (approximately 22-year-old) and healthy human males, who voluntarily donated blood after fasting overnight via sequential ultracentrifugation; The protocol was conducted according to the guidelines of the Declaration of Helsinki, and approved by the Institutional Review Board of Yeungnam University (approval code 7002016-A-2016-021, approval date 4 July 2016).

The density was adjusted appropriately by adding NaCl and NaBr, as detailed elsewhere [56] using a Himac CP100-NX with a fixed angle rotor P50AT4 (Hitachi, Tokyo, Japan) at the LipoLab of Yeungnam University. After centrifugation, each lipoprotein sample was dialyzed extensively against Tris-buffered saline (TBS; 10 mM Tris-HCl, 140 mM NaCl, and 5 mM EDTA [pH 8.0]) for 24 h to remove NaBr.

### 4.7. Purification of Human apoA-I

ApoA-I was purified from HDL by ultracentrifugation, column chromatography, and organic solvent extraction using the method reported by Brewer et al. [57]. At least 95% protein purity was confirmed by SDS-PAGE.

### 4.8. Oxidation of LDL

Oxidized LDL (oxLDL) was produced by incubating the LDL fraction with CuSO_4_ (final concentration, 10 μM for 4 h at 37 °C. OxLDL was then filtered (0.22-μm filter) and analyzed using a thiobarbituric acid reactive substances (TBARS) assay to determine the extent of oxidation with a malondialdehyde (MDA) standard, as described previously [58].

### 4.9. Synthesis of Reconstituted HDL

Reconstituted HDL (rHDL) was prepared using the sodium cholate dialysis method [59] at an initial molar ratio of 95:5:1:0, 95:5:1:0.5, 95:5:1:1, and 95:5:1:2 for POPC:cholesterol:apoA-I:α-syn, respectively. The size and hydrodynamic diameter of the rHDL particles were determined using 8–25% native polyacrylamide gradient gel electrophoresis (PAGGE, Pharmacia Phast System, Uppsala, Sweden) by comparison with the standard globular proteins (GE Healthcare, Uppsala, Sweden).

### 4.10. DMPC Clearance Assay

The interactions of the apoA-I and α-syn with DMPC (dimyristoylphosphatidylcholine) were monitored using a slight modification of the method described by Pownall et al. [60]. The DMPC to protein mass ratio was 2:1 (*w*/*w*) in a total reaction volume of 0.76 mL. The measurements were initiated after adding DMPC and monitored at 325 nm every two minutes using a DU-800 UV-Vis spectrophotometer (Beckman-Coulter, Fullerton, CA, USA) equipped with a thermocontrolled cuvette holder adjusted to 24.5 °C.

### 4.11. Glycation of apoA-I with α-syn

The glycation sensitivity was compared by incubating the purified lipid-free apoA-I (final 1 mg/mL) with 250 mM D-fructose in 200 mM potassium phosphate/0.02% sodium azide buffer (pH 7.4), as reported elsewhere [61]. ApoA-I was incubated for up to 144 h in an atmosphere containing 5% CO_2_ at 37 °C with or without α-syn. The extent of the advanced glycation reactions was determined by reading the fluorescence intensities at 370 nm (excitation) and 440 nm (emission), as described elsewhere [62].

### 4.12. Western Blotting

The glycated mixture of apoA-I (2 µg) and α-syn protein (2 µg) in the lipid-free state was loaded and electrophoresed on 15% SDS-PAGE gels and detected by apoA-I antibody (ab52945, London, UK) as the first antibody (diluted 1:2000) and goat anti-rabbit Immunoglobulin G-horseradish peroxidase (HRP) (A120-101P, Bethyl Laboratories, Montgomery, AL, USA) as the secondary antibody (diluted 1:5000).

### 4.13. Inhibition of LDL Oxidation

The extent of oxidation was determined by incubating the purified human LDL with 10 µM CuSO_4_ for up to 4 h in the presence of α-syn in a lipid-free or lipid-bound state (final 2 µM of protein). The spectroscopic data was verified by subjecting the incubated to electrophoresis on 0.5% agarose gels to compare their electromobilities after visualization of band by Coomassie blue staining. The migration of each lipoprotein depends on its intact charge and size.

### 4.14. Zebrafish

Zebrafish and embryos were maintained using standard protocols. The Committee of Animal Care and Use of Yeungnam University (Gyeongsan, Korea) approved the maintenance of zebrafish and the procedures using zebrafish (YUHS 01-13-004). The fish were maintained in a system cage at 28 °C during treatment under a 12:12 h light cycle with the consumption of plain diet (TetrabitGmbh D49324, Melle, Germany).

### 4.15. Microinjection into Zebrafish Embryos

The embryos at one-day post-fertilization (dpf) were injected individually by a microinjection using a pneumatic picopump (PV830; World Precision Instruments, Sarasota, FL, USA) equipped with a magnetic manipulator (MM33; Kantec, Bensenville, IL, USA) with a pulled microcapillary pipette-using device (PC-10; Narishigen, Tokyo, Japan). The injections were performed at the same position on the yolk to minimize bias. Either α-syn or apoA-I in the lipid-free state were injected into flasks of embryos (final 5 μM, 50 nL). After the injection, live embryos were observed under a stereomicroscope (Motic SMZ 168; Hong Kong) and photographed using a Motic cam2300 CCD camera. (Motic, Hong Kong, China)

### 4.16. Statistical Analysis

All data are expressed as the means ± SD from at least three independent experiments with duplicate samples. The results were compared using a student’s t-test on the SPSS program (version 12.0; SPSS, Inc., Chicago, IL, USA). Statistical significance was defined as *p* < 0.05.

## Figures and Tables

**Figure 1 molecules-26-07485-f001:**
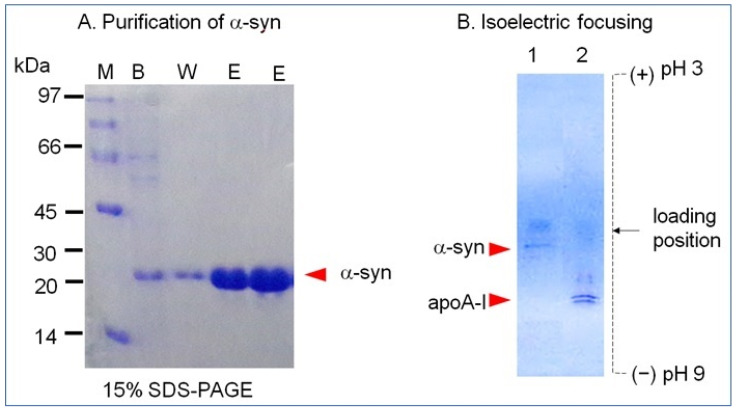
Purification and isoelectric focusing of recombinant human α-synuclein. (**A**) Purification of α-syn using the pET-30a expression system and Ni-NTA column chromatography. Lane 1, binding buffer; lane 2, washing buffer; lane 3, elution buffer; M, molecular weight marker. (**B**) Electromobility on an isoelectric focusing gel. Lane 1, α-syn; lane 2, apoA-I.

**Figure 2 molecules-26-07485-f002:**
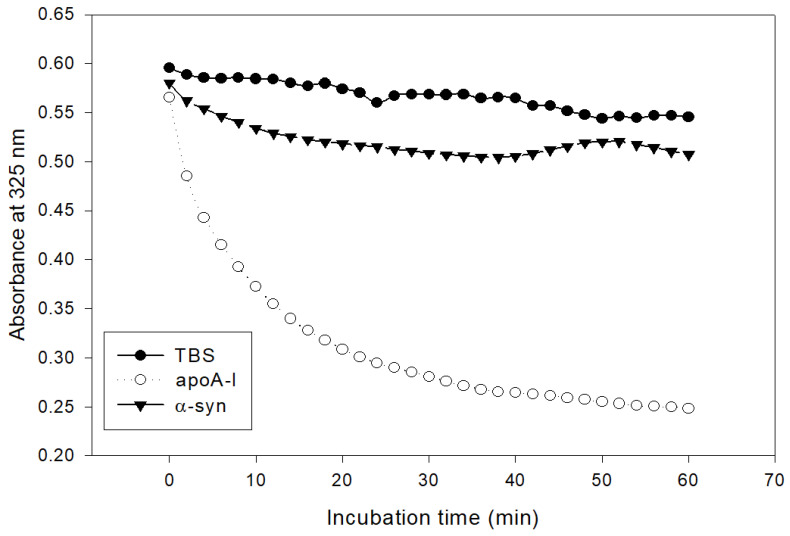
Kinetics of apoA-I and α-syn with DMPC multilamellar liposomes. The absorbance at 325 nm was monitored at 24.5 °C at 2 min intervals.

**Figure 3 molecules-26-07485-f003:**
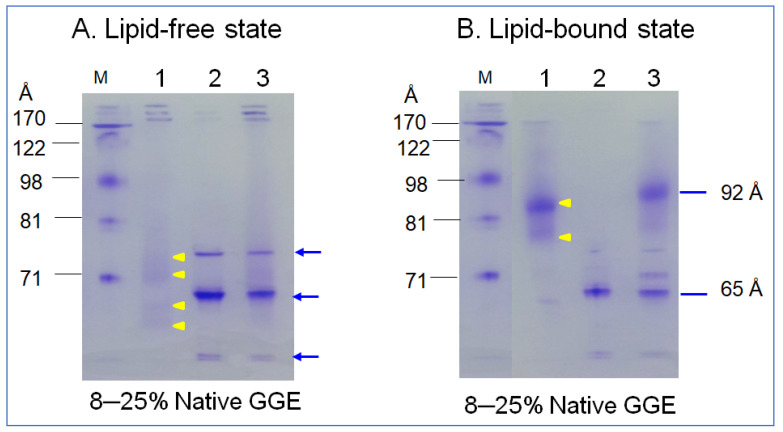
Electrophoretic patterns of apoA-I (lane 1), α-syn (lane 2), and a mixture of apoA-I and α-syn (lane 3) in the lipid-free state (**A**) and lipid–bound state (**B**) under a non-denaturing state in 8–25% gradient gel electrophoresis (GGE). Yellow arrowheads indicate scattered band pattern of apoA-I in lipid-free and lipid-bound state. Blue arrow indicates three distinct bands of α-syn (lane 2) and the mixture of apoA-I and α-syn (lane 3).

**Figure 4 molecules-26-07485-f004:**
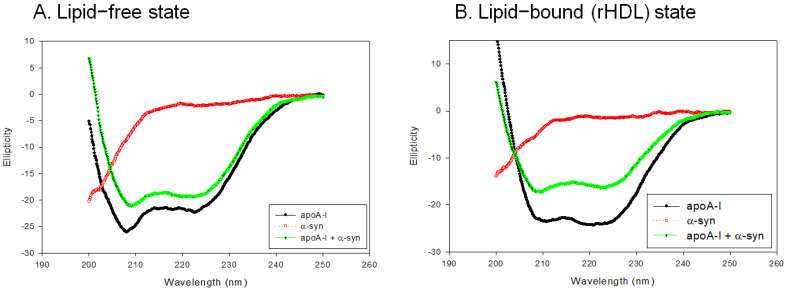
Circular dichroism spectra of apoA-I, α-syn, and a mixture of apoA-I and α-syn in the lipid-free state (**A**) and lipid-bound state (**B**).

**Figure 5 molecules-26-07485-f005:**
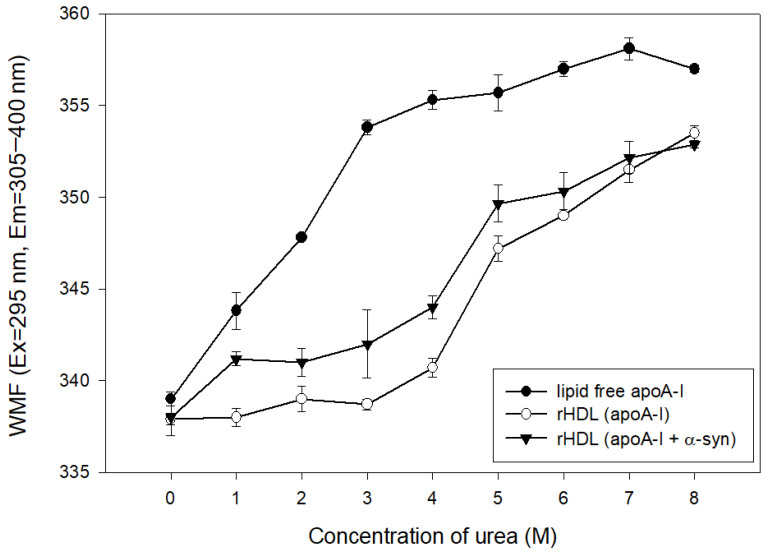
Isothermal denaturation of rHDL containing apoA-I and α-syn. Exposure of Trp fluorescence was detected by fluorospectroscopy (Ex = 295 nm, Em = 310–400 nm).

**Figure 6 molecules-26-07485-f006:**
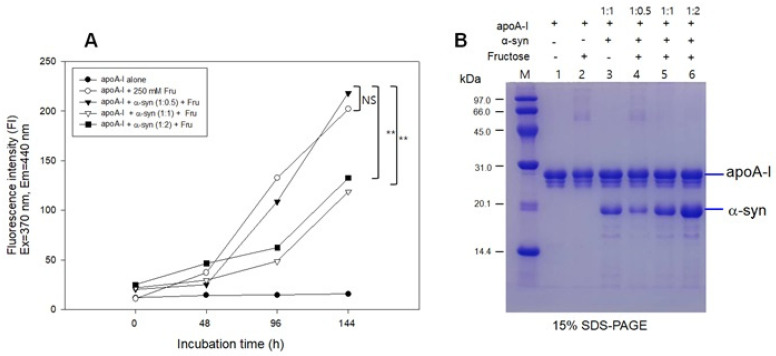
Glycation of apoA-I and α-syn by the fructose treatment. (**A**) Fluorescence determination of advanced glycation end product. (**B**) Electrophoretic patterns of glycated apoA-I and α-syn as visualized by Coomassie Brilliant Blue staining. (-) and (+) indicates absence and presence, respectively. NS, not significant; ** *p* < 0.01.

**Figure 7 molecules-26-07485-f007:**
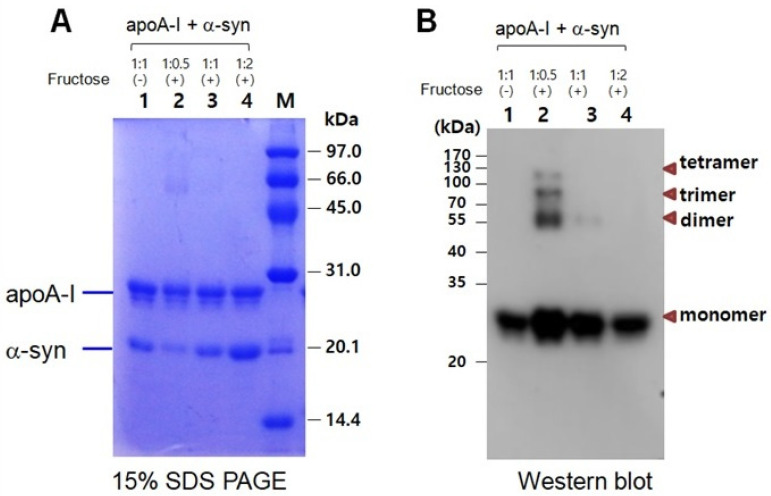
Electrophoretic patterns of a mixture of apoA-I and α-syn after glycation as visualized by Coomassie Brilliant Blue staining (**A**) and immunodetection with apoA-I antibody (**B**). Lane 1, apoA-I:α-syn (1:1) without fructose; lane 2, apoA-I:α-syn (1:0.5) with fructose; lane 3, apoA-I:α-syn (1:1) with fructose; lane 4, apoA-I:α-syn (1:2) with fructose. (-) and (+) indicates absence and presence, respectively.

**Figure 8 molecules-26-07485-f008:**
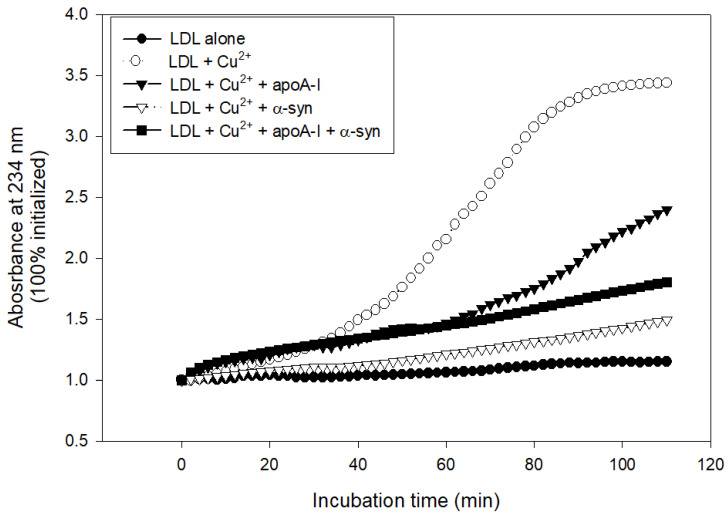
Conjugated diene production of LDL as the absorbance of 234 nm in the presence of cupric ions, apoA-I, and α-syn.

**Figure 9 molecules-26-07485-f009:**
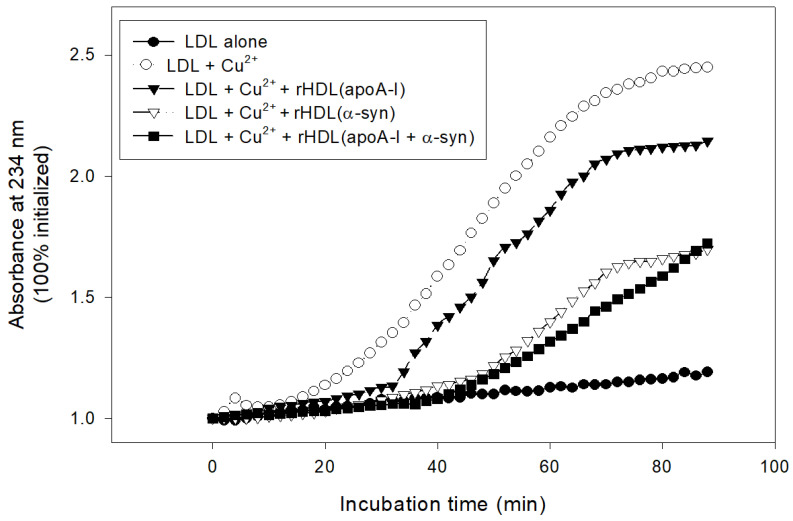
Inhibitory activity of rHDL containing apoA-I and α-syn against cupric ion mediated LDL oxidation. The extent of conjugated diene production was monitored at absorbance at 234 nm.

**Figure 10 molecules-26-07485-f010:**
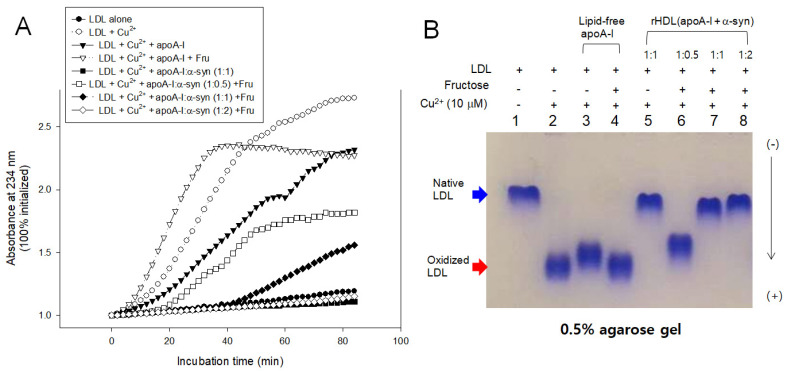
Inhibitory activity of apoA-I and α-syn against LDL oxidation in the presence of fructose (final 250 mM). (**A**) Continuous monitoring of conjugated diene production by absorbance at 234 nm during 90 min. (**B**) Electrophoretic mobility of oxidized and glycated LDL in the presence of apoA-I and α-syn in the rHDL. (-) and (+) indicates absence and presence, respectively.

**Figure 11 molecules-26-07485-f011:**
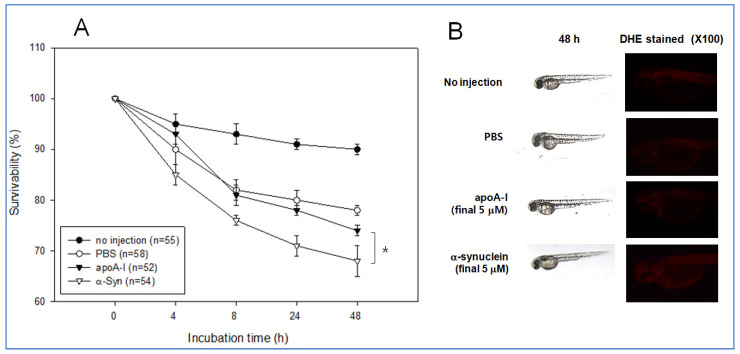
Comparison of physiological toxicity between apoA-I and α-syn in zebrafish embryos. (**A**) Survivability of embryos during 48 h incubation. * *p* < 0.05 (**B**) Representative image of the developmental status of the embryo and visualization of ROS by DHE staining.

**Table 1 molecules-26-07485-t001:** Secondary structure characterization of apoA-I and α-syn in either the lipid-free state and lipid-bound state.

	Lipid-Free			Lipid-Bound (POPC:FC:apoA-I: α-syn = 95:5:1:1)		
	α-helix Content	WMF	Isoelectric Point (pI)	α-helix Content	WMF	Particle Size (Å)
apoA-I alone	56	341	6.4	76	337	86, 78
α-syn alone	-	-	4.5	-		65
apoA-I + α-syn	51	332	-	45	339	92, 74, 69, 65

α-syn, α-synuclein; FC, free cholesterol; POPC, 1-palmitoyl-2-oleoyl-sn-glycero-3-phosphocholine; WMF, wavelength maximum fluorescence.

## Data Availability

The data used to support the findings of this study are available from the corresponding author upon reasonable request.

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
