# Peer review of "Structural and Functional Changes of Reconstituted High-Density Lipoprotein (HDL) by Incorporation of α-synuclein: A Potent Antioxidant and Anti-Glycation Activity of α-synuclein and apoA-I in HDL at High Molar Ratio of α-synuclein"

_molecules, 2021, doi:10.3390/molecules26247485_

Round 1

Reviewer 1 Report

This interesting study investigates interactions between alpha-synclein and apoA1 and their impact on structural and functional characteristics of HDL. The study is well designed, and performed analyses offers a novel insight into possible contribution of both proteins and HDL in the pathogenesis of Parkinson’s disease. Although this research is limited to apoA1 and alpha-synclein and does not include other protein constituents of HDL it still provides novel information that could be important for better understanding of the role of HDL and its structural alterations in various diseases.

The Discussion section should be extended and the authors should address the issue of possible clinical relevance of their findings for diagnosis, as well as for the therapy of Parkinson’s disease. Also, the possible interaction of alpha-synclein with other important HDL-associated proteins (apoE, paraoxonase-1, PAF-AH, S1-P, SAA..) should be briefly discussed and previous results should be mentioned, if any.

Minor comments:

Figures 3 and 7: It would be more clear and easier for the reader if the figure legend would contain an explanation what is represented in each line

Title of paragraph 2.5 is uncompleted

Author Response

Dear Reviewer 1:

Thanks a lot for helpful comments

Please find attached file as response 

Reviewer 2 Report

The manuscript of Cho concerning the association of alpha-synuclein, apoA-I and lipids is an interesting work of some significance.

However, the manuscript is difficult to follow.

for example:

abstract 11-12: “The alpha-Syn and that purified using the pET30a expression vector and E.coli expression system was expressed to elucidate the physiological effects of alpha-Syn on lipoprotein metabolism”.

introduction 43-44: “Parkinson’s disease (PD) is the second most neurodegenerative disorder”

While there are some description about alpha-SYN in the introduction, I feel more information is needed about apoA-I. In the discussion, it is mentioned that “there is a high sequence homology between alpha-syn and apoA-I” (line 277). It would be better to be mentioned in the introduction.

The paragraph (line 69-76) about viral infection is also a little confusing.

What does DMPC stand for?

It is also a little surprising that the first paragraph of the results describes alpha-Syn purification in detail, it may be omitted.

The author mentioned that “addition of alpha-Syn causes the aggregation of apoA-I in the mixture” (line 120). Can there be aggregation at the conditions of CD and fluorescence assays?

Based on the title and the abstract, the manuscript focuses on the interaction of alpha-Syn, apoA-I and rHDL. It is stated in the introduction that “brain cholesterol handling is strongly dependent on the high-density lipoprotein (HDL) metabolism, and HDL-like particles can be synthesized from the brain and glial cells [11]. In the brain, there are no LDL-like particles and apo-B containing lipoproteins as cholesterol carriers [12].” However, after detailed investigation, there is a paragraph in the Results (2.7) on Inhibition of LDL oxidation.

What is the reason for the tissue regeneration assay? Does alpha-synuclein is expressed in tail fin physiologically? The author states that “These results suggest that alpha-syn exhibited adequate beneficial activity in the embryos and tissues of vertebrates without significant toxicity.” (line 258-259). However, in Fig. 12, there are significant differences between PBS and apoA-I as well as between apoA-I and alpha-Syn, but not between PBS and alpha-Syn.

Author Response

Dear Reviewer 

Thanks a lot for your helpful comments.

Please find attached files for the response and changes as per your comments

Round 2

Reviewer 2 Report

The author has adequately answered my questions.

Author Response

Thank a lot for your consideration.

Your comments were helpful to improve this paper